# Ultrasound Stratification of Hepatic Steatosis Using Hepatorenal Index

**DOI:** 10.3390/diagnostics11081443

**Published:** 2021-08-10

**Authors:** Stephen I. Johnson, Daniel Fort, Kenneth J. Shortt, George Therapondos, Gretchen E. Galliano, Theresa Nguyen, Edward I. Bluth

**Affiliations:** 1Department of Radiology, Ochsner Clinic Foundation, 1514 Jefferson Hwy., New Orleans, LA 70121, USA; theresa.nguyen@ochsner.org (T.N.); ebluth@ochsner.org (E.I.B.); 2Center for Outcomes Research, Ochsner Clinic Foundation, 1514 Jefferson Hwy., New Orleans, LA 70121, USA; daniel.fort@ochsner.org; 3Department of Internal Medicine, Northside Hospital Gwinnett, Lawrenceville, GA 30046, USA; shortt.kenneth@gmail.com; 4Multi-Organ Transplant Institute, Ochsner Clinic Foundation, 1514 Jefferson Hwy., New Orleans, LA 70121, USA; gtherapondos@ochsner.org; 5Department of Pathology, Ochsner Clinic Foundation, 1514 Jefferson Hwy., New Orleans, LA 70121, USA; ggalliano@ochsner.org; 6Ochsner Clinical School, The University of Queensland Faculty of Medicine, 1514 Jefferson Hwy., New Orleans, LA 70121, USA

**Keywords:** hepatorenal index, nonalcoholic fatty liver disease, steatosis, ultrasound, liver fat quantification, hepatology, HRI

## Abstract

Hepatorenal index (HRI) has been shown to be an effective, noninvasive ultrasound tool to screen patients for those with or without >5% hepatic steatosis. Objective: The aim of this study was to further refine this HRI tool in order to stratify patients according to their degree of liver steatosis and give direction as to which patients should undergo random liver biopsy. Methods: We conducted a retrospective review of 267 consecutive patients from 2015 to 2017 who had abdominal ultrasounds and a subsequent random liver biopsy within one month. The HRI was calculated and compared with the percent steatosis as assessed by histology. Results: An HRI of ≤1.17 corresponds with >95% positive predictive value of ≤5% steatosis. Between HRI values 1.18 and 1.39, performance of steatosis prediction is mixed. However, for values <1.37 there is an increased likelihood of steatosis ≤5% and likewise the opposite for values >1.37. An HRI of ≥1.4 corresponds with >95% positive predictive value of ≥10% steatosis. Conclusion: HRI is an accurate noninvasive tool to quantify degree of steatosis and guide who should undergo random liver biopsy, potentially significantly reducing the total number of necessary liver biopsies.

## 1. Introduction

Liver steatosis in nonalcoholic patients is a significant public health risk, and its significance cannot be understated. Nonalcoholic fatty liver disease (NAFLD) contributes 75% of the chronic liver disease burden in the Western world [1]. This is particularly worrisome as NAFLD can lead to steatohepatitis, liver fibrosis, and cirrhosis, as well as decompensated liver disease, hepatocellular cancer, and death. NAFLD is also rapidly becoming one of the most common indications for liver transplantation [1].

The majority of patients have a diagnosis of NAFLD made during a nonquantitative assessment of liver fat content at the time of routine ultrasonography and in conjunction with minor elevations in their liver enzymes. The gold standard for the global assessment of NAFLD remains liver biopsy [2] that can assess liver steatosis (fat) as well as inflammatory activity (steatohepatitis) and fibrosis. Serum tests such as interleukin-6 levels potentially could be used in the future to confirm the absence of nonalcoholic steatohepatitis (NASH) [3]. However, currently, they are investigational and not in any kind of routine clinical use.

Liver steatosis can be quantified by imaging modalities. Noninvasive methods include ultrasound, [4] computed tomography (CT), [5] and magnetic resonance imaging (MRI) [6]. While these modalities cannot differentiate microvesicular from macrovesicular fat, they are effective at assessing hepatic fat content [5]. Once proven effective, these noninvasive imaging quantitative methods potentially could be used for assessing steatosis in clinical trials and the suitability of living donors prior to organ transplantation. The controlled attenuation parameter (CAP) at the time of transient elastography (FibroScan^®^, Echosens) is currently the most commonly used method producing a quantitative assessment of liver fat and is widely available although it is operator dependent [7]. Additionally, widely available, MRI has been shown to produce accurate quantitative results. However, both require specialized equipment and incur additional costs to the patient.

A study by Marshall et al. demonstrated that sonographic hepatorenal index (HRI) is a simple and reliable noninvasive method to rule out hepatic steatosis (defined as >5% macrovesicular steatosis) and to avoid unnecessary biopsies [8]. However, this method required additional time and special software to determine HRI. A later study by Shiralkar et al. confirmed that standard, readily available ultrasound equipment can reliably determine steatosis >5% by using HRI determined using DICOM images on PACS and without supplementary software [9] and could therefore be performed as part of the routine liver evaluation without any additional cost. Both previous studies determined HRI values that optimized sensitivity and specificity for the detection of liver steatosis at the 5% level only. The present study used the previously described and proven methodology described by Shiralkar et al. [9] with a larger cohort to determine optimized HRI values for interpretation across the entire range of steatosis to expand the clinical utility of the HRI as a clinical diagnostic tool and ultimately reduce the use of invasive diagnostic testing.

## 2. Materials and Methods

Institutional review board approval was obtained for this study. Because this analysis was retrospective, no informed consent was required. Data were compiled in a manner compliant with the Health Insurance Portability and Accountability Act.

### 2.1. Patient Selection

A medical record review was performed using Epic Systems electronic medical record (Verona, WI, USA) to compile a list of patients who had undergone both an ultrasound-guided liver biopsy and ultrasound of the abdomen from January 2015 to July 2017. By institutional protocol, imaging was always done no more than 30 days before the liver biopsy.

### 2.2. Exclusion Criteria

Patient medical records and ultrasound images were examined for renal disease or abnormalities. Any patients with chronic kidney disease, renal cysts, hydronephrosis, or renal cortical scarring that altered the appearance of the renal cortex were excluded from the analysis. Additionally, patients with significant liver masses or artifact affecting the area where the liver measurements were to be made were excluded.

### 2.3. Sonographic Examination

Sonographic examinations were performed using high-resolution ultrasound (HDI5000 and iU22, Philips Healthcare, Andover, MA, USA) by multiple (n = 12) ultrasound technologists certified by the American Registry for Diagnostic Medical Sonography. Examination types ranged from complete abdominal examinations to limited right upper quadrant examinations. In general, patients received nothing by mouth for at least 8 h before examination. A curved 5-MHz transducer was used, and the image gain parameters were optimized by the technologist, resident, or staff radiologist performing the scan in order to produce optimal visualization of the liver and right kidney. The focal zone was adjusted to be within or deeper than the liver and kidney. To be included in this study, patients had to have a sagittal or oblique image in which both the liver and the right kidney were well visualized. The quality of the ultrasound images was confirmed by a board-certified radiologist at the time of acquisition. Images were archived into a PACS (Impax, versions 6.2.x and 6.3.x, Agfa HealthCare, Greenville, SC, USA).

### 2.4. Image Analysis and HRI Calculation

Once a patient met the criteria for analysis, HRI was calculated using the region of interest (ROI) measure tool standard on PACS software (Agfa IMPAX, edition 6.6.x). The ROI function displays various statistics, including mean brightness and size in pixels. Mean brightness is calculated after the image is obtained using numeric values assigned to grayscale pixels in the ROI, ranging from 0 (black) to 255 (white), respectively. The units of mean brightness are, therefore, arbitrary.

An ROI in the liver was selected, and the measure function was used to produce an average brightness and to verify size (number of pixels >400). An ROI in the kidney was selected at the same depth of field as the one in the liver. The measure function was used to produce an average brightness and to verify size. ROIs in the liver could not include large ducts or vessels, masses, or cysts. ROIs in the kidney could contain only the renal cortex and could not include masses, cysts, collecting system, or any extrarenal tissue. Patients who had sonographic changes of kidney disease such as cortical thinning or loss of differentiation between the cortex and medulla were excluded since an accurate HRI could not be obtained. Portions of the scan that were affected by artifact were excluded, and ROIs were at the level of or superficial to the focal zone. The average liver brightness was divided by the average kidney brightness to produce the HRI. This calculation was repeated at different levels for a total of 3 HRI values that were averaged to produce a final value. All image analyses were performed by radiology residents and supervised by an abdominal imaging fellowship trained attending radiologist with greater than 3 years of experience. All were blinded to the biopsy results. The criteria for an adequate sample required that at least 3 areas of the liver and kidney could be sampled (Figure 1). Average brightness of the liver and right kidney was recorded on a spreadsheet for analysis and comparison.

### 2.5. Histologic Sampling

Radiology residents and staff radiologists performed the biopsies under ultrasound guidance. After sterile preparation, draping, and local anesthesia, a 16-gauge biopsy gun (Monopty; Bard Biopsy Systems, Tempe, AZ, USA) was used to take random samples of hepatic parenchyma. Samples were generally taken from the right hepatic lobe, although not necessarily from the same region where the HRI measurement was obtained. The number of samples was based on the adequacy of the biopsy specimen during the procedure. A board-certified pathologist specializing in liver pathology evaluated all formalin-fixed specimens to visually estimate the percentage of macrovesicular and microvesicular fat in increments of 5%. Pathologic quantification of fat was entered into a spreadsheet for analysis and comparison.

### 2.6. Analysis

Descriptive characteristics for HRI and steatosis for relevant subgroups were tabulated and reported. Positive predictive value (PPV) was chosen for the balancing summary statistic over sensitivity, specificity, J statistic, etc., to better reflect the overall goal of the effort which was to most confidently define the interpretation of an HRI value. Decision points for threshold values (10%, 5% steatosis) were identified as the HRI value where at least 95% PPV was possible. Plots were created to investigate the relationship, distribution, and PPV of data between the HRI and steatosis.

## 3. Results

Originally, 289 patients were verified to have abdominal ultrasound exam and liver biopsy performed within a 1-month duration; however, 12 patients were excluded for liver masses obstructing the hepatorenal view, 4 examinations had insufficient hepatorenal views to reliably calculate HRI, 3 patients had profound medical renal disease, 1 patient was excluded due to sarcoidosis that obscured the hepatorenal view, 1 patient did not have an abdominal ultrasound performed and instead had a liver biopsy performed by another hospital service, and 1 patient was excluded for hydronephrosis (Figure 2). Therefore, 267 patients had both abdominal ultrasound examinations and liver biopsies that met the previously mentioned study criteria.

Of the 267 patients, 121 (45.3%) were male and 146 (54.7%) were female. The mean age was 52.19 years ± 13.13 years. The median age was 53 years, and mode was 51 years. Forty unique conditions were documented as indications for the complete abdominal ultrasound examinations. Of the 267 patients, 238 had only 1 condition documented as the indication for the ultrasound examination, 28 had 2 conditions documented as indications for the ultrasound examination, and 1 patient had 3 conditions documented as indications for the ultrasound examination. No patients had more than 3 conditions listed under the indication for the complete abdominal ultrasound examination. The most common indication among all patients was elevated liver function tests, which was listed as an indication for liver ultrasound in 119 (44.6%) patients (Table 1).

Among the 141 patients with ≤5% steatosis on biopsy, the leading indications for ultrasound examination were elevated liver function tests (44.0% of patients), hepatitis C (16.3%), hepatitis B (9.9%), liver mass not obstructing the hepatorenal view (6.4%), hyperbilirubinemia (4.3%), and unspecified cirrhosis (3.5%) (Table 1).

The leading indications for liver ultrasound for the 3 patients with 6–9% steatosis are not categorized because the numbers are so small, but are listed in Table 1.

Among the 123 patients with ≥10% steatosis, the most common indications for liver ultrasound were elevated liver function tests (46.3%), hepatitis C (12.2%), NAFLD (8.9%), NASH (5.7%), liver mass not complicating the hepatorenal window (5.7%), hepatitis B (4.9%), hepatic steatosis (4.1%), cirrhosis (3.3%), hepatocellular carcinoma (HCC) screening (2.4%), and hyperbilirubinemia (2.4%)(Table 1).

The 267 charts were reviewed again by a staff radiologist. Four records were referred to pathology for repeat examination of the percent steatosis and were verified to be correct, producing 267 records that met the stated criteria for calculating the HRI.

### 3.1. HRI Measurements

For the 267 patients in this study, the mean HRI was 1.40 ± 0.46 (median 1.28, range 0.61–3.19). The mean HRI for patients with ≤5% steatosis was 1.09 ± 0.25 (n = 141). Patients with steatosis ≥10% had a mean HRI of 1.76 ± 0.56 (n = 123). Only 3 measured values were classified as between 6% and 9% steatosis and were excluded from summary analysis (Figure 3).

### 3.2. HRI Decision Point Analysis

The effectiveness of HRI for the diagnosis of steatosis ≤5% and ≥10% was evaluated against histologic sampling results. Our analysis was based on the PPV of each clinical decision (steatosis ≤5% vs. ≥10%) at every observed value of HRI. An HRI ≤ 1.17 denotes a sensitivity and specificity of 64% and 97%, respectively, with a 95% PPV for steatosis ≤5%. HRI ≥ 1.4 indicates a sensitivity and specificity of 4% and 99%, respectively, with a 95% PPV for steatosis ≥10%. (Figure 3). Figure 3 also demonstrates a visualization of PPV across observed HRI values. An HRI of 1.18–1.39 is an indeterminant range in which patients may undergo biopsy if clinically indicated; however, an HRI of 1.18–1.37 indicates likely ≤5% steatosis, and an HRI of 1.38–1.39 indicates likely ≥10% steatosis. Based on the pathologic correlation, 72% (193/267) of observations fall into ranges where the PPV for steatosis prediction is ≥95%.

## 4. Discussion

It has been shown that steatosis can be identified with ultrasound due to increased echogenicity of the fatty liver. However, ultrasound has been limited and faulted because of the technical expertise required, resulting in interobserver variability in both the production of images and their interpretation. With this study, we expanded the work of Marshall et al. and Shiralkar et al. [8,9] and demonstrated HRI can be used to accurately stratify patients into the following categories: ≤5% steatosis with 95% specificity who do not need random liver biopsy to quantify steatosis, likely ≤5% steatosis for whom biopsy may be beneficial, likely >5% steatosis for whom biopsy may be beneficial, and ≥10% steatosis who do not need random liver biopsy to quantify hepatic steatosis. Our findings suggest that a significant number of patients will not need to undergo biopsy to access steatosis. Up to 72% of patients in our study could have avoided a biopsy to access for degree of >95% PPV. This quantitative assessment should provide greater reliability and assurance of accurate ultrasound interpretation. The high PPV of the decision should also result in greater clinical utility and adoption of this technique by hepatologists and other clinicians. By using the HRI, we can now diagnose the presence of hepatic steatosis without the need of liver biopsy in many patients.

### 4.1. Accuracy of HRI and Decision Points

These results represent an analysis of HRI with the largest sample size and most accurate instrumentation to date. Shiralkar et al. reported that an HRI of 1.34 or higher had 92% sensitivity for identifying fat exceeding 5%, an 85% specificity, a 94% negative predictive value, and a 79% PPV [9]. In this work, new decision points for ≤5% (HRI ≤ 1.17) and ≥10% (HRI ≥ 1.4) steatosis were chosen based on a threshold of 95% PPV with the goal of eliminating the need for liver biopsy within these thresholds. Using these guidelines, the outcomes of 72% of observed biopsies (193/267) could have been predicted with a PPV of >95%. For 28% of HRI values between the thresholds of 1.18 and 1.39, predictions as to the more likely pathologic outcome can still be made. However, biopsy may still be indicated based on patient presentation and clinical judgment. Repeated measures to establish HRI trends or correlation with other biomarkers may be able to resolve remaining uncertainty. To that end, we will follow this study with an analysis of HRI and steatosis trend data on the same patient population when repeat measures become available due to continued surveillance.

### 4.2. Evaluation of Steatosis by Other Modalities

Evaluation of hepatic steatosis has been studied extensively by CT and MRI. On CT, steatosis can be assessed as hypoattenuation in relation to the attenuation of the spleen as an internal reference, which has a sensitivity of 88% to 95% and specificity of 90% to 99% on noncontrast CT [10]. Absolute liver attenuation of ≤40 Hounsfield units has been shown to be the most accurate method of determining moderate to severe steatosis (histologic grade ≥ 30%) [11]. While CT can effectively evaluate steatosis, the radiation risk of CT makes it less appealing as a screening modality compared to ultrasound or MRI. Multiple MR techniques can be used for the detection of hepatic steatosis such as in-phase/out-of-phase imaging utilizing the Dixon method, MRI imaging with and without fat saturation, magnetic resonance spectroscopy, and proton density fat fraction [12,13]. All of these techniques offer noninvasive, sensitive, and specific tools to assess for hepatic steatosis; however, MRI poses difficulties of cost, accessibility, and acquisition parameters [14].

CAP is a relatively recent test that is usually done at the same time as vibration-controlled transient elastography (FibroScan^®^) [15] Its validity has been demonstrated in large studies with decision points of >10%, >33%, or >66% steatosis [16,17]. However, this methodology uses proprietary equipment which cannot be readily used for other imaging purposes.

### 4.3. Applications in Hepatology

The most obvious application in hepatology is confirmation that a patient with suspected NAFLD truly has hepatic steatosis. Although conventional ultrasound is frequently suggestive of steatosis, confirmation using HRI is useful and does not require additional resources. Furthermore, quantification with HRI should allow us to trend the degree of steatosis over time which may be particularly relevant in evaluating treatment of steatosis. Considering the noninvasive nature of this methodology, HRI could also be used routinely to follow patients at regular intervals to determine if preventive measures are effective in reducing or preventing steatosis in high risk patients. This tool also can be used to routinely monitor treatment responses to patients diagnosed with steatosis after they have undergone appropriated treatments.

### 4.4. Limitations

Accurate HRI assessment requires experience and appreciation of the technical requirements of diagnostic ultrasound. These have been previously described by Marshall et al. and Shiralkar et al. [8,9]. In the current study, in more than 92% of patients, an HRI assessment can be successfully obtained and numerical number calculated even when using a large number of technologists to perform the studies. HRI cannot be reliably used in patients with chronic renal disease, absent right kidney, or liver masses adjacent to the right kidney. Accurate assessment also requires visualization of the liver parenchyma without interference by structures such as hepatic cysts, portal veins, or the gallbladder in the area of measurement. A single oblique image showing both the right kidney and liver must be obtained for the ratio to be accurate. While challenging, the methodology can be easily learned and is reproducible. In this study, only 7.3% (21/289) of patients were excluded because of the inability to technically adequately obtain an accurate HRI assessment. This level of technical success was achieved even though many more technologists participated in performing the imaging studies and calculations than were involved in the work of Marshall et al. and Shiralkar et al. [8,9]. In fact, over the course of the three years of the study, a considerable number of the technologists were newly hired and had to be initially trained to apply the HRI measurement tool. The low number of technical exclusions in this study demonstrates that new technologists can be trained to accurately obtain an HRI measurement even when they are unfamiliar with the methodology. All measurements for this study were made using one manufacturer’s proprietary equipment. Other equipment could have different decision point numbers; therefore, this work should be reproduced before the decision point values are adopted for equipment manufactured by other vendors. Another limitation to this study is that a cost benefit analysis of using the ultrasound HRI methodology compared to MR or CT was not performed. This analysis needs to be done in the future and we expect that it will show that the use of an ultrasound technique to measure steatosis would be economically preferable to other methodologies. In 28% of patients with HRI calculations between 1.18–1.39 (75/267), the PPV is not at the 95% range and therefore further assessment may still be required. Nevertheless, with the HRI value calculated for these patients, we can still strongly suggest, depending on the specific value, as described in this paper, what the degree of steatosis is. However, if greater clinical certainty is required for any of the members of this group, a biopsy could be obtained for greater accuracy. This would still result in a significant reduction in the number of interventional biopsy procedures currently performed. Perhaps the most significant limitation and disappointment of this study is the inability to identify other definitive decision markers for ranges of >10% steatosis. We were unsuccessful in identifying accurate >20% or >30% steatosis decision markers. Perhaps this could be corrected if a larger number of subjects were enrolled in a future study who have greater degrees of steatosis than in our study population.

### 4.5. Key Take Home Message

By utilizing the HRI, we can noninvasively and accurately determine quantifiable ranges of steatosis with standard ultrasound equipment. Seventy-two percent (72%) of our observations fall into ranges where the positive predictive value for steatosis prediction is ≥95%, and as a result, the total number of necessary liver biopsies could potentially be significantly reduced. This test can also be used to accurately follow changes in steatosis over time to determine if therapeutic interventions are successful and if disease is progressing or regressing.

## 5. Conclusions

We have shown that we can accurately determine quantifiable ranges of steatosis with standard ultrasound equipment. Seventy-two percent (72%) of our observations fall into ranges where the PPV for steatosis prediction is ≥95%. We can separate ≤5% and ≥10% steatosis in the liver without the need for a biopsy. An HRI ≤ 1.17 denotes ≤5% steatosis and an HRI ≥ 1.4 indicates ≥10% steatosis. For values between 1.18–1.37, there is an increased likelihood of steatosis ≤5%. For values 1.38–1.39, there is an increased likelihood of ≥10% steatosis. Because this methodology is noninvasive and free of side effects, this test can be used to accurately follow changes in steatosis over time to determine if therapeutic interventions are successful and if disease is progressing or regressing.

## Figures and Tables

**Figure 1 diagnostics-11-01443-f001:**
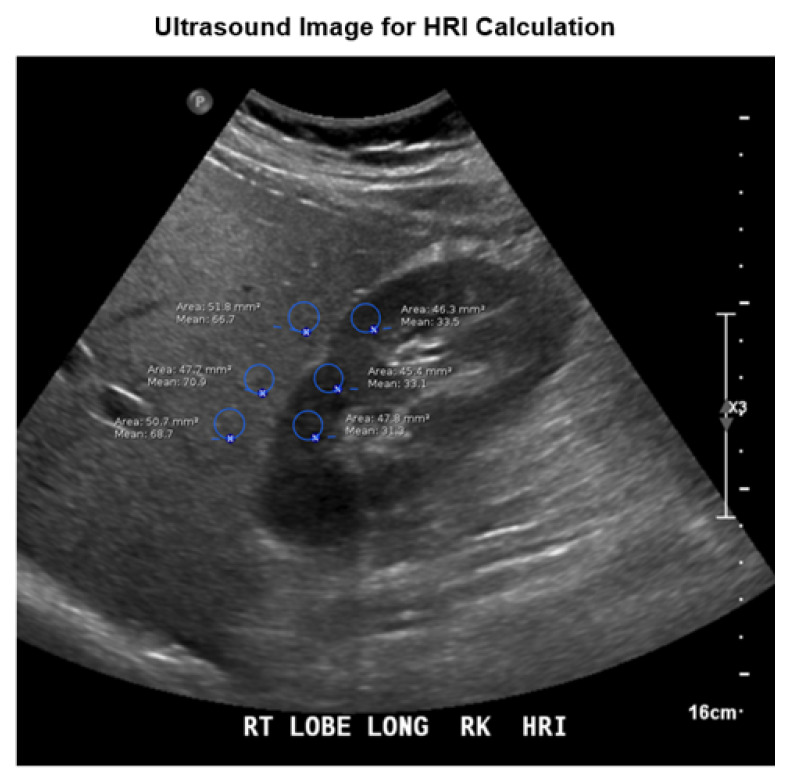
Example of HRI measurements. HRI is determined by placing region of interest markers at the same depth in the kidney and the adjacent liver parenchyma.

**Figure 2 diagnostics-11-01443-f002:**
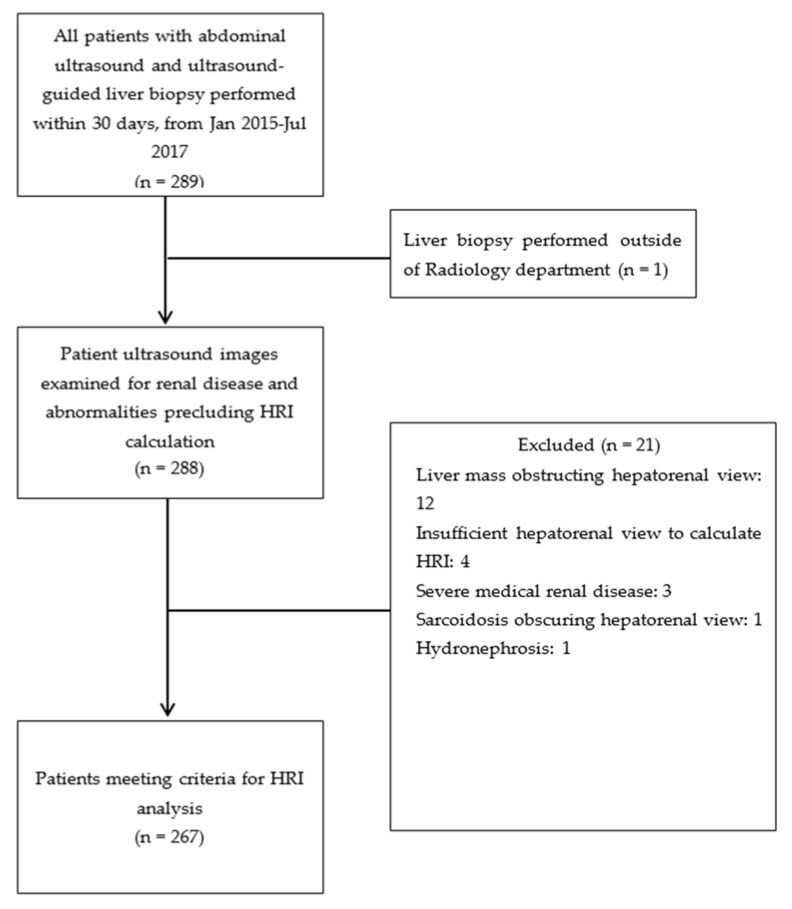
Diagram illustrating patient selection and exclusion. Abbreviation: HRI, hepatorenal index.

**Figure 3 diagnostics-11-01443-f003:**
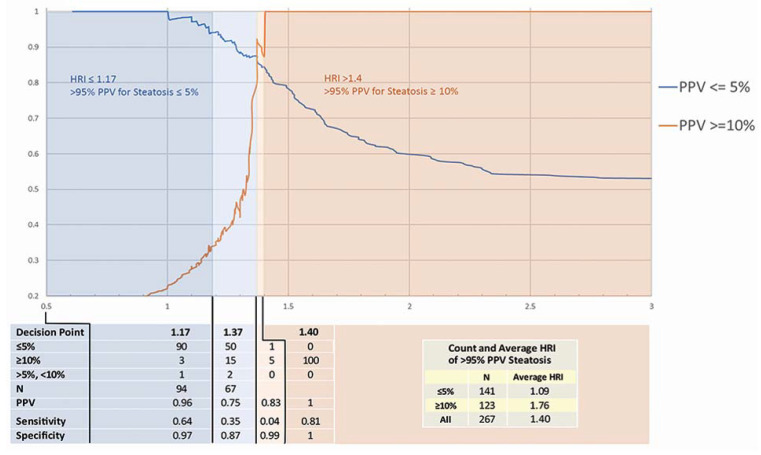
Positive predictive value for steatosis based on HRI.

**Table 1 diagnostics-11-01443-t001:** Patient demographics and exam indications.

	Parameter	Number	Percent
**Sex**			
	Male	121	45.3%
	Female	146	54.7%
**Age (average)**		52.19 ± 13.13	
	**Indication**	**Number of Patients with Indication Listed**	**Percent of Subjects with Indication Listed**
**Patients with ≤5% steatosis**		(Total 141 patients)	
	elevated liver function tests	62	44.0%
	hepatitis C	23	16.3%
	hepatitis B	14	9.9%
	liver mass	9	6.4%
	hyperbilirubinemia	6	4.3%
	unspecified cirrhosis	5	3.5%
	liver failure	3	2.1%
	autoimmune hepatitis	3	2.1%
	liver disease (unspecified)	3	2.1%
	Other indications (i.e., abdominal pain, cholangitis)	≤ 2	≤1.4%
**Patients with ≥10% Steatosis**		(Total 123 Patients)	
	elevated liver function tests	57	46.3%
	Hepatitis C	15	12.2%
	NAFLD ^1^	11	8.9%
	NASH ^2^	7	5.7%
	liver mass	7	5.7%
	hepatitis B	6	4.9%
	hepatic steatosis	5	4.1%
	cirrhosis	4	3.3%
	HCC ^3^ screening	3	2.4%
	hyperbilirubinemia	3	2.4%
	Other indications (i.e., jaundice, elevated ALP ^4^)	≤2	≤1.6%
**Patients with >5% to <10% steatosis**		(Total 3 patients)	
	hepatitis B	1	
	HCC ^3^ screening	1	
	RUQ ^5^ pain	1	
	hepatosplenomegaly	1	

^1^ Nonalcoholic fatty liver disease; ^2^ nonalcoholic steatohepatitis; ^3^ hepatocellular carcinoma; ^4^ alkaline phosphatase; ^5^ right upper quadrant.

## Data Availability

The data presented in this study are available on request from the corresponding author. The data are not publicly available due to HIPPA compliance.

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
