# Peer review of "Ultrasound Stratification of Hepatic Steatosis Using Hepatorenal Index"

_diagnostics, 2021, doi:10.3390/diagnostics11081443_

Round 1

Reviewer 1 Report

General comments

This study evaluates the hepatorenal index as a noninvasive tool to screen patients with hepatic steatosis lower or greater than 5%. Overall, the analysis is interesting but could be further expanded by reporting the sensitivity, specificity, and accuracy. The description of indications of ultrasound could be significantly reduced. Conclusions regarding the replacement of liver biopsy should be toned down.

Specific comments

Abstract:

-Conclusion: “potentially reducing the total number of necessary liver biopsies by more than 70%”. How do the Authors derive this data? I suggest to tone down the conclusion on liver biopsy since this is often need to assess inflammation and fibrosis.

Introduction:

-Page 1, lines 32-33: “Nonalcoholic fatty liver disease (NAFLD) contributes 75% of the chronic liver disease burden”. I suggest to specify that this is referred to Western countries.

Materials and methods:

-Image analysis, page 3: Please specify the years of experience of the readers involved in imaging analysis. As the analysis was performed on PACS and the HRI was not prospectively calculated at the time of ultrasound exams, this could represent a significant limitation for this study due to the know inter-reader variability and inter-observer difference on ultrasound examination.

-Histologic sampling: Please specify other cutoff assessed in the histopathological analysis and which reference was used to calculate hepatic steatosis.

-Please provided at least one imaging example of HRI measurements in the methods.

Results:

-Page 4, lines 145-153: I suggest to avoid using “record” instead of patients in the exclusion criteria.

-Page 6-8: I think that the number and type of conditions and indications for abdominal ultrasound could be irrelevant for the purpose of this study. I suggest to remove the description of conditions (except for the most common one) for the main text and to simplify Table 1. There is no need to repeat indications in both main text and table 1.

-Page 8, lines 242-243: “The mean HRI for patients with ≤5% steatosis was 1.09 ± 0.25 (n=141). Patients with steatosis ≥10% had a mean HRI of 1.76 ± 0.56 (n=123)”. Was there a statistically significant difference?

-Page 8, lines 243-244: “Only 3 measured values were classified as between 6% and 9% steatosis and were excluded from summary analysis”. It is unclear why the Authors decided to exclude these patients for the summary analysis as the purpose of this study was to evaluate the performance for the diagnosis of 5% hepatic steatosis. Why than the cutoff was placed at 10%? It is suggested to not to exclude these cases as 5% was the established reference standard for the histopathological diagnosis of steatosis.

-While optimal cutoff for rule out or rule in hepatic steatosis with high positive predictive value are important, it should be also important to know the sensitivity, specificity and accuracy of a diagnostic test. This could have been easily calculated for this study. Consider to report it.

Discussion:

-I suggest to tone down the conclusion regarding the need of liver biopsy since this is often needed to diagnose inflammation and fibrosis. Contrarily, these results may be relevant to monitor the evolution of hepatic steatosis after appropriate therapies. This may be discussed in depth.

Author Response

All comments are attached and additions/revisions are included in the updated version of the manuscripts.

Reviewer 2 Report

The present paper has high clinical value. I strongly believe it advances the knowledge in the field.

Could you discuss the impact of obesity (measured by BMI ) on obtaining reliable results?

Is HRI correlated with the BMI?

Thank you!!

Author Response

We did not look into BMI relationship of HRI.  That is an area of future research.

Round 2

Reviewer 1 Report

Thank you for considering all the comments and suggestions and for providing the revisions accordingly. I have not further comments for this manuscript.